# DiNO-Diffusion: Scaling Medical Diffusion Models via Self-Supervised Pre-Training

## Abstract

Diffusion models (DMs) require large annotated datasets for training, limiting their applicability in medical imaging where datasets are typically smaller and sparsely annotated. We introduce DiNO-Diffusion, a self-supervised method for training DMs that conditions the generation process on image embeddings extracted from DiNO, a pretrained vision transformer. By not relying on annotations, our training leverages over 868k unlabelled images from public chest X-Ray (CXR) datasets. DiNO-Diffusion shows comprehensive manifold coverage, with FID scores as low as 4.7, and emerging properties when evaluated in downstream tasks, allowing to generate semantically-diverse synthetic datasets even from small data pools, demonstrating up to 20% AUC increase in classification performance when used for data augmentation. Results suggest that DiNO-Diffusion could facilitate the creation of large datasets for flexible training of downstream AI models from limited amount of real data, while also holding potential for privacy preservation. Additionally, DiNO-Diffusion demonstrates zero-shot segmentation performance of up to 84.4% Dice score when evaluating lung lobe segmentation, evidencing good CXR image-anatomy alignment akin to textual descriptors on vanilla DMs. Finally, DiNO-Diffusion can be easily adapted to other medical imaging modalities or state-of-the-art diffusion models, allowing large-scale, multi-domain image generation pipelines for medical imaging.

## 1 Introduction

Diffusion models (DMs) have recently emerged as robust and proficient foundational models in medical imaging, exhibiting substantial capabilities in image generation, image enhancement, reconstruction, and segmentation (Kazerouni et al., 2023). The field of synthetic image generation in particular has greatly shifted to text-to-image DMs, generating images that are nearly indistinguishable from real ones (Osorio et al., 2024; Chambon et al., 2022a; Ye et al., 2023; Aversa et al., 2023; Pinaya et al., 2022) and facilitating remarkable zero-shot performance in segmentation and classification tasks (Tian et al., 2023; Zhang et al., 2023a). However, DMs depend on the availability of large datasets containing images paired with corresponding descriptors (usually text) to guide the generation process, a requirement that presents a considerable obstacle in the medical domain (Beddiar et al., 2023). Medical imaging datasets are typically small, contain free-form and inconsistent annotations including captions, binary labels or segmentations, and are generally prohibitively costly to compile and curate (Beddiar et al., 2023). To address these challenges, some works have proposed pseudo-labeling with vision-language models (VLMs; Betker et al. (2023)) or have trained lean mapping networks over frozen pretrained backbones to reduce the number of required annotated samples (Li et al., 2023; Zhang et al., 2023b). However, despite their promise, pseudo-labelling approaches find limited applicability in the medical field given a lack of high-quality medical imaging captioners (Beddiar et al., 2023). In addition, while some authors have successfully trained mapping networks to bridge the gap between unimodal foundation models, they still require relatively large annotated datasets to be trained (Beddiar et al., 2023).

These limitations represent important roadblocks for medical DMs. While the natural imaging literature focuses on saturating generation quality by improving the base architecture, optimization process or condition alignment (Esser et al., 2024; Betker et al., 2023; Liu et al., 2024), the medical imaging community navigates these hurdles by leveraging smaller or custom-annotated datasets (Chambon et al., 2022a; Ye et al., 2023; Osorio et al., 2024; Aversa et al., 2023; Pinaya et al., 2022).

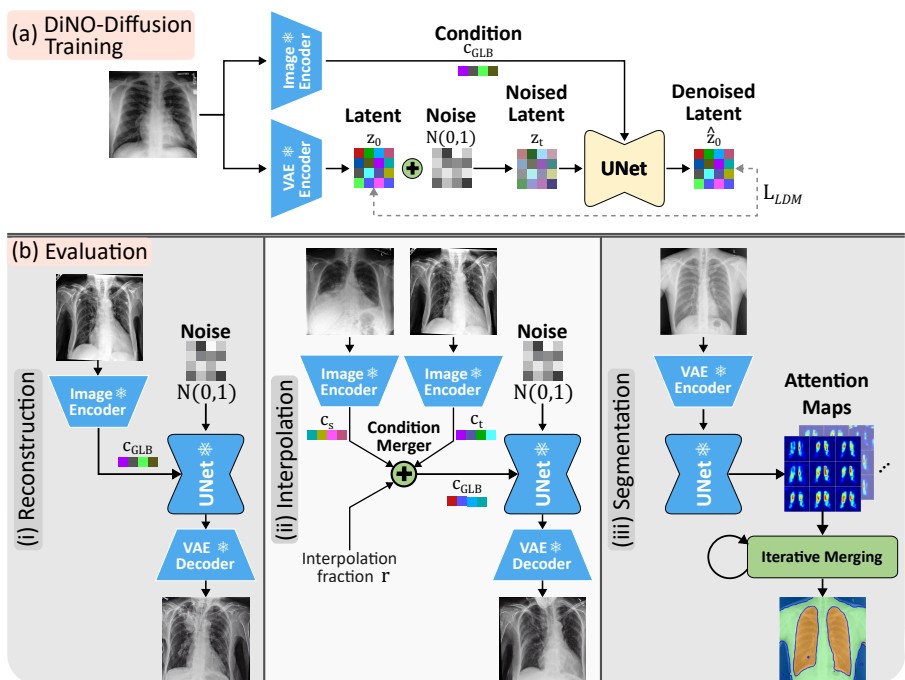

Figure 1: DiNO-Diffusion's training (a) and evaluation (b) protocols. (a) the training image is both embedded into latents $z_0$ with a frozen (❄) VAE, and processed by a frozen image encoder to generate global tokens that act as condition $c_{GLB}$. Then, the latents are noised at timestep $z_t$ and fed along the condition to the UNet, which denoises the latent $\hat{z}_0$. Then, the loss $L_{LDM}(z_0, \hat{z}_0)$ is computed. (b) the trained UNet is used to produce: (b-i) "reconstructions" of a given image; (b-ii) "interpolated" synthetic images from the embeddings of a source ($c_s$) and a target ($c_t$) real images at interpolation fraction $r$; or (b-iii) segmentations, by iteratively merging latent attention maps.

Moreover, although mapping networks have found their footing in the diffusion literature with approaches such as ControlNet (Zhang et al., 2023b), these would still rely on large-scale medical DMs trained with prohibitively extensive amounts of annotated images. In this context, applying a self-supervised approach to DM training would be highly beneficial for medical image synthesis. Self-supervision enables models to learn from unlabelled data, providing exceptional results in multiple downstream tasks when used as image embedders (Caron et al., 2021; Oquab et al., 2023; Pérez-García et al., 2024; Dippel et al., 2024; Moutakanni et al., 2024).

With that in mind, we introduce DiNO-Diffusion, a novel self-supervised methodology for training medical DMs at scale which conditions the image generation process on image-derived tokens extracted from a frozen DiNO model (Caron et al., 2021; Oquab et al., 2023), as opposed to textual descriptors. DiNO-Diffusion allows independence from existing annotations, circumventing the limitations imposed by the scarcity and inconsistency of medical image labels. Moreover, it is agnostic to the choice of DM architecture, medical imaging modality or optimization strategy. To test this, a model was trained on a large corpus of open-source CXR data found in the literature which do not share any common labeling or descriptor required to train regular DMs (e.g., text captions), achieving low FID scores and high image quality. DiNO-Diffusion can generate medical images despite using DiNO embeddings, which are derived from natural images. To test the alignment between DiNO embeddings and generated images, several downstream evaluation tasks were performed, comprising classification and segmentation, which addressed the model's ability to improve classification performance when adding synthetic data to a pool of real data or when fully replacing real with synthetic data; and assessing whether a self-supervised DM can be used to create zero-shot segmentation masks for distinct anatomical structures.

In summary, our main findings are as follows: (1) DiNO-Diffusion allows training large DMs given its independence from specific architectures, imaging modalities, available annotations, dataset sizes or optimization strategies. (2) DiNO's embeddings are descriptive enough for image generation de-

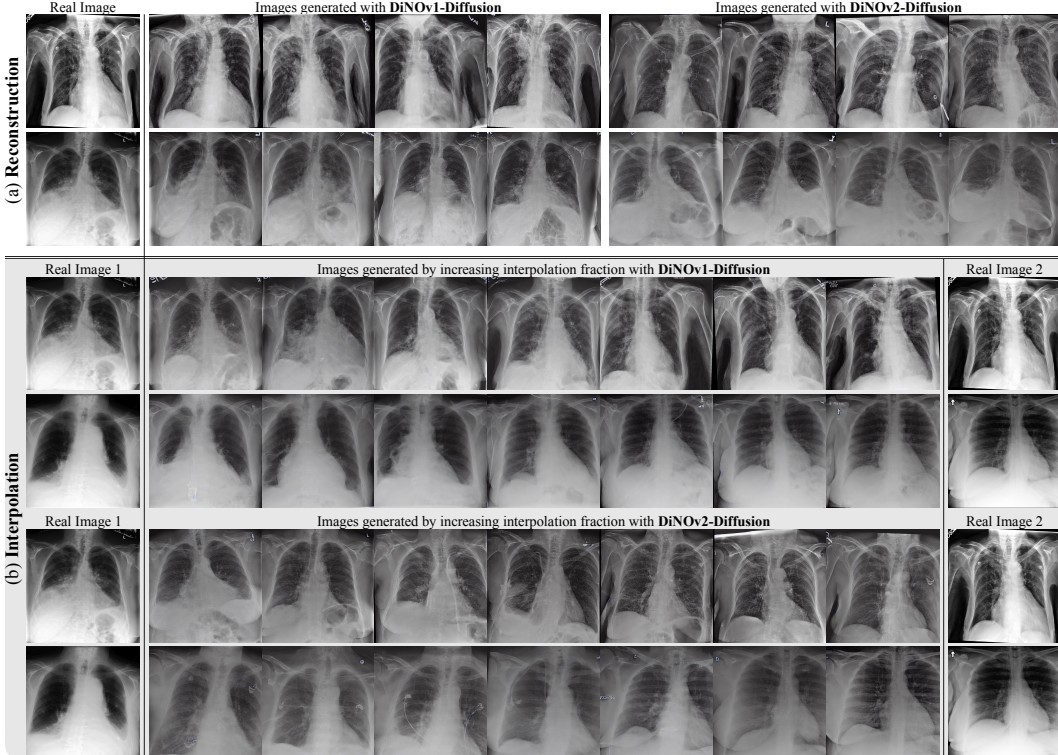

Figure 2: Examples of generated images with DiNO-Diffusion. In the reconstruction experiment (a), each row represents randomly generated examples from two base images within MIMIC and for both DiNOv1- and DiNOv2-Diffusion, showing semantic variability. In the interpolation experiment (b), each row depicts two real images and the result from generating synthetic images by interpolating the embeddings incrementally for DiNOv1-Diffusion (b-top) and DiNOv2-Diffusion (b-bottom).

spite not being trained on medical images. Using DiNO's global tokens seemed to bottleneck enough information to introduce semantic variability during DiNO-Diffusion's generation, thus avoiding replication of the input data. (3) DiNO-Diffusion was used to generate semantically-diverse synthetic datasets even from small data pools. These samples were used for data augmentation, improving classification performance on different data regimes. In addition, training on only synthetic data showed potential for mitigating privacy concerns. (4) DiNO-Diffusion can be leveraged for zero-shot medical image segmentation through iterative attention map merging. This demonstrates its ability to learn semantic coherence and its good alignment with anatomic structures. To our knowledge, this is the first application of zero-shot segmentation applied to medical DMs.

## 2 METHODS

This Section explains the methodology employed for studying the self-supervised DM. In Section 2.1, the datasets used for training and evaluation are described. In Section 2.2, the model's architecture and theoretical background is outlined. In Section 2.3, the designed mechanisms for self-supervised conditioning are detailed. In Section 2.4, the evaluation protocol employed to benchmark model performance is defined. Finally, in Section 2.5, the specific parameters used for model training and evaluation are enumerated. Figure 1 visually describes the training and evaluation pipeline.

### 2.1 DATA

To explore DiNO-Diffusion's self-supervision capability, a large-scale dataset comprised of every openly accessible CXR dataset found in the literature (de la Iglesia Vayá et al., 2023; Irvin et al., 2019; Goldberger et al., 2000; Johnson et al., 2019; Demner-Fushman et al., 2015; Tabik et al.,

2020; Jaeger et al., 2014; Candemir et al., 2014; Bustos et al., 2020; Cohen et al., 2021; Reis et al., 2022; Shiraishi et al., 2000; Kermany et al., 2018; Cohen et al., 2020; Chowdhury et al., 2020; Rahman et al., 2021; JF Healthcare, 2020; Nguyen et al., 2022; Pham et al., 2023; Zawacki et al., 2019; Liu et al., 2020; Rahman et al., 2024; Fedorov et al., 2021)[1] was collected, reaching over 1.2M total images from 21 distinct data providers. Three different subsets were taken from this compound dataset for different purposes. Firstly, a subset comprising every dataset minus MIMIC-CXR (Johnson et al., 2019) was selected for training the DiNO-Diffusion models. Their labels were discarded and label balancing was not performed, resulting in 868 394 samples with a variety of image sources, resolutions and patient characteristics. Secondly, MIMIC-CXR was used solely for evaluating the model via two classification tasks (see Section 2.4). MIMIC-CXR is composed of chest radiographs with free-text radiology reports, for which multi-label classification information is available. The MIMIC-CXR dataset was preprocessed to match similar literature (Chambon et al., 2022a) by discarding lateral views, by restricting the labels to those whose prevalence was of at least 4% (Atelectasis, Cardiomegaly, Consolidation, Edema, Pleural Effusion, Pneumonia and Pneumothorax), and by splitting its p10-p18 subsets for classifier training and leaving p19 as a held-out test set. Finally, the third subset for the segmentation task relied on three small datasets containing annotated masks: the JSRT ($N = 247$), Montgomery ($N = 138$) and Shenzhen ($N = 663$) datasets (Shiraishi et al., 2000; Jaeger et al., 2014).

## 2.2 GENERATIVE ARCHITECTURE - STABLE DIFFUSION

Latent Diffusion Models (LDMs) approach image generation as an iterative denoising process, transforming pure noise $x_T$ into a defined image $x_0$ over $T$ steps with a parameterized DM $\epsilon_\theta(z_t, t, c)$, where $c$ represents an optional condition. LDMs address the prohibitive computational demands of traditional DMs by reducing the dimensionality of the input. LDMs currently find active development with ongoing research in different parameterised models, optimization strategies and dimensionality reduction pipelines.

This study adopts the Stable Diffusion (SD) framework (version 1, Rombach et al. (2022)) as its baseline. Despite being outperformed by more recent models and its output size limitation of 512x512 pixels, SD's lightweight architecture, open-source nature, and community adoption makes it ideal for our proof of concept. SD comprises a frozen *variational autoencoder* (VAE) and a trainable *conditional denoising UNet*.

The VAE consists of an encoder ($\mathcal{E}$) and a decoder ($\mathcal{D}$). The encoder compresses fixed-size images $x \in \mathbb{R}^{H \times W \times 3}$ into a latent $z = \mathcal{E}(x) \in \mathbb{R}^{(H/d) \times (W/d) \times k}$, where $k = 4$ is number of channels extracted by the VAE and $d = 8$ is the downsampling factor. The decoder maps latents back to the original image space $\hat{x} = \mathcal{D}(z)$. Stable Diffusion's VAE has been shown to generalize to medical data (Chambon et al., 2022a;b). The UNet serves as the diffusion component and uses a ResNet architecture as its convolutional backbone, where the condition $c$ is incorporated through attention mechanisms (see Section 2.3).

With this model, training with conditional information involves two phases: the *forward* and *reverse diffusion* processes. During the *forward* diffusion, an image $x_0$ (or its latent representation $z_0$) and condition $c$ are chosen. A timestep $t$ is randomly selected ($t \sim \mathcal{U}(1, ..., T)$) so a noisy latent $z_t$ is generated by mixing $z_0$ with noise $\epsilon \sim \mathcal{N}(0, 1)$, resulting in a *partially noised* latent. The *reverse* process uses the UNet to estimate the original noise $\epsilon$ from $z_t$, $t$ and $c$.

The network is optimized using the Mean Squared Error (MSE) loss between the predicted and actual noise to adjust the weights of the UNet:

$$\mathrm{L}_{LDM} = \mathrm{E}_{z \sim \epsilon(x),\, c,\, \epsilon \sim \mathcal{N}(0,1),\, t} \left[ ||\epsilon - \epsilon_\theta(z_t, t, c)||_2^2 \right] \tag{1}$$

After training, image synthesis begins with sampling a noisy latent $z_T \sim \mathcal{N}(0, 1)$, progressively denoising it with condition $c$ to obtain $z_0$ so that $\hat{z}_0 = \epsilon_\theta(z_{T:0}, c)$, and by using the VAE's decoder, so that $\hat{x} = \mathcal{D}(\hat{z}_0) = \mathcal{D}(\epsilon_\theta(z_{T:0}, c))$.

---

[1]Thanks, among others, to the National Library of Medicine, National Institutes of Health, Bethesda, MD, USA.

Table 1: AUC scores (mean $\pm$ SD; 5-fold cross-validation) for (a) data augmentation experiments and (b) full synthetic trainings across DiNO-Diffusion variants, image synthesis strategies (reconstruction, interpolation), real-to-synthetic ratios ($rs$) and data regimes ($N$). The baseline (i.e., training with real data only) test performances are depicted at the top in light-blue. **Bold** values represent best performance improvement relative to the real-only baseline for each data regime, DiNO-Diffusion model and synthesis strategy. Asterisks (*) represent statistical significance ($p < 0.05$).

| Strategy | | | $rs$ ratio | $AUC_{N=50}\downarrow$ | $AUC_{N=100}\downarrow$ | $AUC_{N=500}\downarrow$ | $AUC_{N=1000}\downarrow$ | $AUC_{N=5000}\downarrow$ |
|---|---|---|---|---|---|---|---|---|
| Real data | | | 1:0 (real-only) | $0.548 \pm 0.013$ | $0.566 \pm 0.047$ | $0.682 \pm 0.011$ | $0.715 \pm 0.005$ | $0.747 \pm 0.006$ |
| **(a) Data Augmentation** | DiNOv1-Diffusion | Reconstruction | 1:1 | $0.551 \pm 0.037$ | $0.602 \pm 0.025$ | $0.685 \pm 0.012$ | $\mathbf{0.724 \pm 0.002}$ * | $0.756 \pm 0.002$ * |
| | | | 1:5 | $0.564 \pm 0.050$ | $0.626 \pm 0.016$ | $\mathbf{0.706 \pm 0.010}$ * | $0.725 \pm 0.005$ | $\mathbf{0.756 \pm 0.003}$ * |
| | | | 1:10 | $0.608 \pm 0.024$ * | $0.618 \pm 0.030$ | $0.701 \pm 0.014$ | $0.719 \pm 0.007$ | $0.745 \pm 0.012$ |
| | | | 1:50 | $\mathbf{0.650 \pm 0.020}$ * | $\mathbf{0.651 \pm 0.013}$ * | $0.698 \pm 0.009$ | $0.699 \pm 0.012$ | $0.735 \pm 0.006$ |
| | | Interpolation | 1:1 | $0.540 \pm 0.036$ | $0.589 \pm 0.033$ | $0.676 \pm 0.007$ | $0.682 \pm 0.011$ * | $0.686 \pm 0.009$ * |
| | | | 1:5 | $0.579 \pm 0.033$ | $0.625 \pm 0.011$ | $0.696 \pm 0.013$ | $0.706 \pm 0.007$ * | $0.703 \pm 0.007$ * |
| | | | 1:10 | $0.589 \pm 0.039$ * | $0.618 \pm 0.018$ | $\mathbf{0.709 \pm 0.009}$ * | $0.709 \pm 0.003$ | $0.693 \pm 0.018$ * |
| | | | 1:50 | $\mathbf{0.632 \pm 0.015}$ * | $\mathbf{0.644 \pm 0.014}$ * | $0.702 \pm 0.013$ | $\mathbf{0.716 \pm 0.013}$ | $0.743 \pm 0.004$ |
| | DiNOv2-Diffusion | Reconstruction | 1:1 | $0.515 \pm 0.026$ | $0.566 \pm 0.015$ | $0.692 \pm 0.022$ | $0.716 \pm 0.008$ | $\mathbf{0.747 \pm 0.003}$ |
| | | | 1:5 | $0.552 \pm 0.036$ | $0.608 \pm 0.035$ | $\mathbf{0.705 \pm 0.004}$ * | $0.714 \pm 0.006$ | $0.744 \pm 0.004$ |
| | | | 1:10 | $0.611 \pm 0.010$ * | $0.631 \pm 0.029$ | $0.705 \pm 0.006$ * | $\mathbf{0.717 \pm 0.005}$ | $0.745 \pm 0.006$ |
| | | | 1:50 | $\mathbf{0.617 \pm 0.018}$ * | $\mathbf{0.627 \pm 0.016}$ * | $0.700 \pm 0.016$ | $0.710 \pm 0.005$ | $0.744 \pm 0.004$ |
| | | Interpolation | 1:1 | $0.574 \pm 0.043$ | $0.603 \pm 0.049$ | $\mathbf{0.685 \pm 0.009}$ | $0.698 \pm 0.007$ * | $0.681 \pm 0.011$ * |
| | | | 1:5 | $0.580 \pm 0.018$ * | $0.594 \pm 0.053$ | $0.657 \pm 0.023$ | $0.688 \pm 0.011$ * | $0.710 \pm 0.008$ * |
| | | | 1:10 | $0.608 \pm 0.025$ * | $0.622 \pm 0.026$ | $0.681 \pm 0.017$ | $0.694 \pm 0.005$ * | $0.689 \pm 0.021$ * |
| | | | 1:50 | $\mathbf{0.618 \pm 0.020}$ * | $\mathbf{0.649 \pm 0.016}$ * | $0.690 \pm 0.024$ | $0.703 \pm 0.008$ * | $0.702 \pm 0.013$ * |
| **(b) Full Synthetic Training** | DiNOv1-Diffusion | Reconstruction | 1:1 | $0.546 \pm 0.017$ | $0.571 \pm 0.046$ | $0.667 \pm 0.008$ * | $0.696 \pm 0.010$ * | $0.730 \pm 0.004$ * |
| | | | 1:5 | $0.574 \pm 0.059$ | $0.610 \pm 0.029$ * | $\mathbf{0.701 \pm 0.007}$ | $\mathbf{0.724 \pm 0.004}$ * | $0.752 \pm 0.005$ |
| | | | 1:10 | $0.625 \pm 0.020$ * | $0.631 \pm 0.025$ * | $0.701 \pm 0.010$ | $0.722 \pm 0.005$ | $\mathbf{0.753 \pm 0.006}$ |
| | | | 1:50 | $\mathbf{0.655 \pm 0.015}$ * | $\mathbf{0.645 \pm 0.011}$ * | $0.689 \pm 0.018$ | $0.709 \pm 0.014$ | $0.746 \pm 0.006$ |
| | | Interpolation | 1:1 | $0.515 \pm 0.029$ | $0.491 \pm 0.033$ | $0.530 \pm 0.035$ * | $0.546 \pm 0.016$ * | $0.538 \pm 0.020$ * |
| | | | 1:5 | $0.525 \pm 0.015$ * | $0.576 \pm 0.037$ | $0.686 \pm 0.011$ | $0.695 \pm 0.004$ * | $0.531 \pm 0.009$ * |
| | | | 1:10 | $0.572 \pm 0.023$ * | $0.574 \pm 0.013$ | $0.701 \pm 0.005$ * | $0.706 \pm 0.005$ * | $0.686 \pm 0.005$ * |
| | | | 1:50 | $\mathbf{0.635 \pm 0.018}$ * | $\mathbf{0.644 \pm 0.015}$ * | $\mathbf{0.705 \pm 0.013}$ * | $0.711 \pm 0.011$ | $0.736 \pm 0.007$ |
| | DiNOv2-Diffusion | Reconstruction | 1:1 | $0.509 \pm 0.025$ * | $0.564 \pm 0.044$ | $0.646 \pm 0.021$ * | $0.649 \pm 0.005$ * | $0.711 \pm 0.004$ * |
| | | | 1:5 | $0.523 \pm 0.019$ * | $0.591 \pm 0.048$ | $0.684 \pm 0.009$ | $0.700 \pm 0.007$ * | $0.728 \pm 0.007$ * |
| | | | 1:10 | $0.574 \pm 0.034$ | $0.610 \pm 0.021$ * | $0.687 \pm 0.012$ | $0.695 \pm 0.011$ * | $0.730 \pm 0.006$ * |
| | | | 1:50 | $\mathbf{0.603 \pm 0.033}$ * | $\mathbf{0.626 \pm 0.015}$ * | $\mathbf{0.699 \pm 0.014}$ * | $0.708 \pm 0.006$ | $0.741 \pm 0.006$ |
| | | Interpolation | 1:1 | $0.546 \pm 0.045$ | $0.567 \pm 0.019$ | $0.553 \pm 0.035$ * | $0.558 \pm 0.015$ * | $0.533 \pm 0.024$ * |
| | | | 1:5 | $0.536 \pm 0.030$ | $0.593 \pm 0.040$ | $0.631 \pm 0.029$ * | $0.669 \pm 0.008$ * | $0.646 \pm 0.016$ * |
| | | | 1:10 | $0.551 \pm 0.030$ | $0.602 \pm 0.035$ | $0.668 \pm 0.017$ | $0.660 \pm 0.014$ * | $0.680 \pm 0.017$ * |
| | | | 1:50 | $\mathbf{0.610 \pm 0.052}$ * | $\mathbf{0.625 \pm 0.009}$ * | $0.672 \pm 0.018$ | $0.677 \pm 0.005$ * | $0.714 \pm 0.016$ * |

## 2.3 Self-Supervised Conditioning

LDMs condition image generation using a semantic tensor $c$ to guide the diffusion process. This tensor is usually obtained from a frozen transformer model $f_\Phi$ that maps the label information into a tensor $c = f_\Phi(x) \in \mathcal{R}^{S \times N}$, where $S$ is the token length (of variable size), $N$ is the embedding dimension and $x$ represents whichever input the embedder model requires (text, image, etc.). Although the current diffusion literature has mainly focused on using textual descriptors as their main conditioning strategy, other conditioning mechanisms have been employed (Aversa et al., 2023; Pinaya et al., 2022; Zhang et al., 2023b).

In this work we explore conditioning using image-derived semantic descriptors. Specifically, a vision transformer trained with the DiNO method (Dosovitskiy et al., 2021; Caron et al., 2021) was used to produce a semantic description of the image to be generated. Vision transformers split an image into small patches (usually $P = 14px^2$ or $P = 16px^2$) representing "visual words" and operate over them using a standard transformer architecture. The model outputs a tensor of tokens $c = f_\Phi(x) \in \mathcal{R}^{S \times N}$ comprising a class token $c_{CLS} \in \mathcal{R}^N$, sometimes a pooler token $c_{PLR} \in \mathcal{R}^N$, sometimes a predefined amount $R$ of register tokens $c_{REG} \in \mathcal{R}^{R \times N}$ (Darcet et al., 2024), and finally a series of $L$ patch tokens $c_{LCL} \in \mathcal{R}^{L \times N}$, where $L = H/P_y * W/P_x$. Finally, the conditioning tensor outputted by the embedder was reduced to the available global information $c_{GLB} = [c_{CLS}, c_{PLR}, c_{REG}]$ before feeding it to the UNet, as upon initial exploration the patch tokens contained too much local information of the original image $x$ and led to trivial models that learnt to reconstruct images from redundant information (see Section A.1). Figure 1-(a) visually describes the training pipeline.

Conditioning image generation on image embeddings offers flexibility on generation as long as a conditioning embedding exists. In this work, two simple generation strategies were explored, to

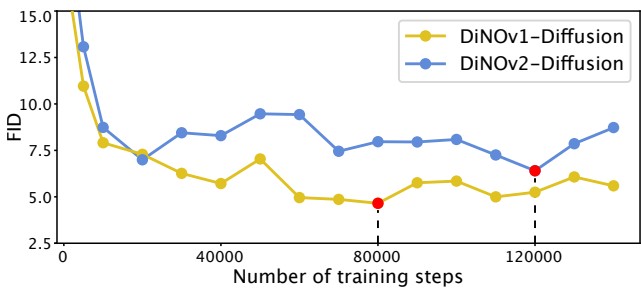

Figure 3: FID scores over a MIMIC subset for DiNO-Diffusion every 2500 steps. Lower is better.

evaluate the model's in-distribution and out-of-distribution performance, although more advanced approaches could be devised:

**Reconstruction-based image generation** the "reconstruction" strategy consists in synthesizing images $\hat{x} = \mathcal{D}(\epsilon_\theta(z_{T:0}, c))$ from the global information of an existing real example $(x, y)$, where $y$ is the image's label, $\hat{y} = y$ and $f_\Phi(x)$ is the conditioning embedding as produced by DiNO. This reconstruction leverages DiNO-Diffusion's large-scale pretraining to produce semantic variations over the source image $x$. Exact replicas of $x$ are prevented by design due to conditioning with the compressed information from DiNO's global embedding, causing a bottleneck. Figure 1 (b-i) depicts the reconstruction process.

**Interpolation-based image generation**: the "interpolation" strategy uses the same image generation mechanism from above. The difference lies in the sampling method of the conditioning embedding $c$, which is interpolated from two images $(x_1, y_1), (x_2, y_2)$ so that $\hat{c} = lerp(f_\Phi(x_1), f_\Phi(x_2), r)$, where $r \in [0, 1]$ is the interpolation fraction. This strategy attempts to generate synthetic images from less sampled regions of the real data manifold, located between existing samples, following approaches such as MixUp (Zhang et al., 2018). See Figure 1 (b-ii) for a visual depiction of this strategy.

## 2.4 EVALUATION

This section details four different evaluation protocols used for benchmarking DiNO-Diffusion:

**Image Quality & Checkpoint Selection**: the Fréchet Inception Distance (FID; Heusel et al. (2017)) was used to quantify generation quality at multiple checkpoints for both variants of DiNO-Diffusion. The FID scores computed for the data generated via the "reconstruction" strategy (see Section 2.3) were used as a proxy for overall model performance. Similarly to Chambon et al. (2022a), FID scores were computed over a 5k subset of MIMIC-CXR's p19 dataset (Johnson et al., 2019), and are reported every 2500 steps in Figure 3. Also following the same work, the FID score was computed on the feature space of a pretrained domain-specific image encoder from TorchXrayVision (Cohen et al., 2022) as opposed to the default Inception-V3 model, as the latter might not provide an accurate measure of image quality when dealing with medical image data. Finally, the optimal checkpoint for each DiNO-Diffusion model was the checkpoint with the lowest FID score.

**Data Augmentation**: this experiment explored DiNO-Diffusion's ability to enhance the sample size of a dataset by training a classification model on real and synthetic data using five-fold cross-validation and testing on a held-out test set (MIMIC's p19). For this purpose, MIMIC's training dataset (p10-p18) was subset into different data regimes with decreasing sample size $\mathcal{X}_n$, with $n \in \{10k, 5k, 1k, 500, 100, 50\}$ samples in the subset. Given that MIMIC has multi-label annotations, label balancing was performed by randomly selecting $n/card(\mathcal{L})$ elements of each label in the labelset $\mathcal{L}$ from $\mathcal{X}$ without replacement, ensuring sufficient representativity of all labels within the training set. Smaller subsets were also enforced to be contained into bigger ones, so that $\mathcal{X}_{\mathcal{N}_{i+1}} \in \mathcal{X}_{\mathcal{N}_i}$. With $\mathcal{X}_n$ defined, synthetic data was created to increase sample size by generating partially-synthetic datasets $\hat{\mathcal{X}}_n$ with real-to-synthetic ratios of 1:1, 1:5, 1:10 and 1:50 for the reconstruction- and interpolation-based synthesis (see Section 2.3). For the reconstruction experiments, ratios larger than 1:1 represent several semantic variations of a single source image $(x, y)$, which aim at introducing realistic variance into the synthetic data while retaining the label-specific

image features. The interpolation experiment addressed whether intermediate embeddings could still be decoded into an image that retains label-specific features from both elements in the pair. For this purpose, the sample pairs were enforced to have at least one label in common (see Section 2.3) without repetition. When not all the labels are in common between the pair, the labels of the interpolated example are set to the ones of the sample it is closest to, as defined by the interpolation fraction $r$. Finally, in the case of not having enough unique pairs for a given split, some pairings were repeated with different $r$.

**Full Synthetic Training**: this experiment explores whether test-set AUC drops when training a classifier solely on synthetic data, to address whether DiNO-Diffusion can serve as a privacy-preserving synthetic replacement for real data. The generation strategies, data regimes, real-to-synthetic ratios and 5-fold cross-validation settings from Section 2.4 were followed as evaluation strategy.

**Zero-Shot Segmentation**: this experiment investigates the model's ability to learn semantic coherence by generating segmentation masks from the internal representations generated during the DiNO-Diffusion's UNet forward pass. For this purpose, the zero-shot segmentation approach from DiffSeg (Tian et al., 2023) was followed, consisting of leveraging the self-attention weights from each transformer block of the UNet and iteratively merging them based on their Kullback-Leibler divergence. This methodology was applied both to DiNO-Diffusion and a vanilla SD model to generate lung lobe segmentation masks without further training. Using a combined dataset of 1,048 cases with ground truth annotations (See Section 2.1), candidate masks were evaluated by their Dice score and selected via non-maximum suppression. The relevant hyperparameters (merging threshold, timestep) as well as the best performing checkpoint were selected per model (see Section B.3). Refer to Figure 1 (b-iii) for a visual depiction of the segmentation pipeline.

## 2.5 EXPERIMENTAL SETUP

The models were trained by adapting HuggingFace Diffusers' script for training DMs (von Platen et al., 2022). The DMs were trained for 100 epochs ($\sim$ 140000 steps) using 4 H100 GPUs per model, an aggregated batch size of 512 ($bs = 64$, gradient accumulation of 2 steps), 8-bit Adam optimizer with constant $lr = 10^{-4}$ and 1000-step warmup and xformers' memory-efficient attention (Lefaudeux et al., 2022). The specific versions of the DiNOv1 and DiNOv2 image encoder architectures used were "`facebook/dino-vitb16`" (Caron et al., 2021) and "`timm/vit-base-patch14-reg4-dinov2`" (Darcet et al., 2024), respectively. The webdataset library (WebDataset Contributors (2021)) was used for storing and streaming data directly from the bucket during all model trainings. The classification experiments were based on training HuggingFace's implementation of a "`densenet121`" for 150 max epochs using T4 GPUs with batch size 64, AdamW optimizer with $lr = 10^{-4}$ and weight decay of $10^{-5}$, a LR reduction-on-plateau scheduler with patience 10 and early stopping after 25 epochs with no validation AUC improvement. For the checkpoint evaluation, a pretrained "`densenet121-res224-all`" (Cohen et al. (2022)) was employed as feature extractor. All images followed the same minimal preprocessing strategy before training or evaluation, similar to other works in the literature (Cohen et al., 2022; Chambon et al., 2022a). Dynamic intensity values (uint8, uint16) were rescaled to uint8. Images were center-cropped with a 1:1 aspect ratio, resized to 512x512 pixels and padded areas were removed. Minimal data augmentations were applied during all model trainings, including random sharpening and affine transformations (5% shearing, 5% translation, 90%-140% scaling).

## 3 RESULTS

**Image Quality & Checkpoint Selection**: the FID scores were calculated every 2500 steps over a subset of MIMIC's p19 dataset following Chambon et al. (2022a). Both the DiNOv1 and DiNOv2 models converged relatively late, reaching scores of 4.7 and 6.4 at 80k and 120k steps, respectively. The full FID scores for every checkpoint can be observed in Figure 3. DiNOv1-Diffusion leads to lower FID scores when compared to DiNOv2-Diffusion. This is also evident by a slightly less saturated synthetic images generated with DiNOv2-Diffusion when compared to the source real images (see Figure 2). Additional generated examples are provided in Section A.2.

**Data Augmentation**: in this experiment, real and synthetic data were used in different proportions to train DenseNet-121 classification models. Table 1-a and Figure 4-a provide the results of the

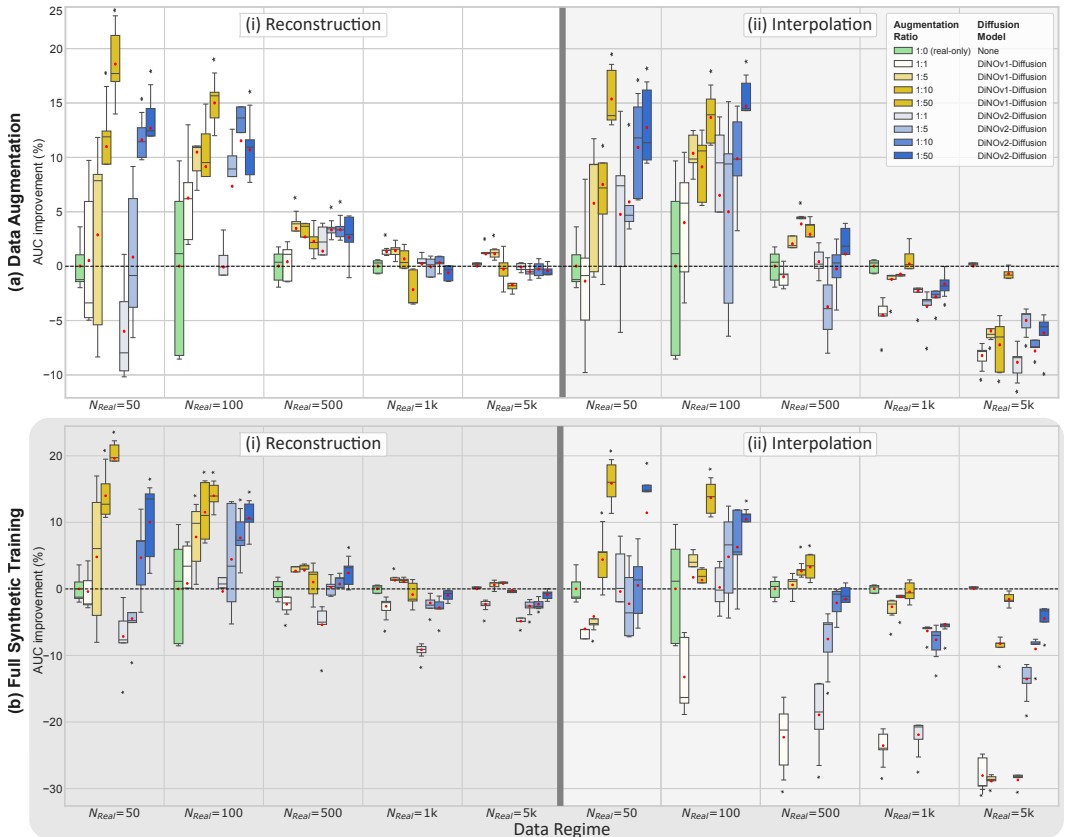

Figure 4: Performance improvements for the Data Augmentation (a) and the Full Synthetic Training (b) experiments. The horizontal line represents a 0% improvement over the mean (red dot) classification performance when using real-data only (green bars) for each data regime and real-to-synthetic ratio ($rs$) independently. Values above the dotted line represent performance improvement. The vertical lines separate the different data regimes for easier comparison, where the performance of DiNOv1-Diffusion (yellow palette) and DiNOv2-Diffusion (blue palette) are jointly displayed. In (i), the results for the reconstruction experiment are explored, whereas (ii) depicts the results for the interpolation experiment. Asterisks (*) represent statistical significance to real baseline ($p < 0.05$).

cross-validation trainings. The 'reconstruction' workstream (see Section 2.3) depicts consistent improvements when used for data augmentation in all data regimes, with AUC increases up to approximately 20% in small-data regimes. In some larger-data regimes ($N \in [1000, 5000]$), the addition of large amounts of synthetic data slightly degraded performance, although never by a significant margin ($p > 0.05$). The 'interpolation' workstream (see Section 2.3) also depicts improvements in smaller data regimes as compared to not using synthetic data, although it leads to a significant performance degradation in large-data regimes ($p < 0.05$). Also, DiNO-Diffusion using DiNOv1 yields larger performance improvements compared to when using DiNOv2. This is always true for both image synthesis strategies, except for the interpolation results on data regime $N_{real} = 100$, where the best test AUC is achieved with DiNOv2 for 1:50 $rs$ ratio.

**Full Synthetic Training**: the test set results of the full synthetic trainings are shown in Table 1-b and Figure 4-b. The data synthesised via the "reconstruction" strategy (see Section 2.3) using DiNOv1-Diffusion provided good performance in almost all settings, where statistically significant performance decreases only existed for the lowest $rs$ ratio in the largest three data regimes. For both "reconstruction" DiNO-Diffusion variants, training with sufficiently large $rs$ ratios in small-data regimes ($N_{real} \in [50, 100, 500]$) led to significant performance improvements of up to 20%, mirroring the data augmentation results (see Section 2.4). However, for the "interpolation" based synthesis (see Section 2.3), this was only the case in the 1:50 ratio. Generally, the data synthesised via the "interpolation" strategy did not reliably train the classifier in splits larger than $N_{real} = 1k$

Table 2: Segmentation performance, measured by mean Dice scores (%). The displayed values are based on the hyperparameter configurations that led to best overall results.

| Dataset | Stable Diffusion 1.5 | DiNOv1-Diffusion | DiNOv2-Diffusion | Fully Supervised |
|---|---|---|---|---|
| Threshold | 0.5 | 0.05 | 0.05 | - |
| Timestep | 300 | 300 | 300 | - |
| Grid size | 32x32 | 16x16 | 16x16 | - |
| Shenzhen | $80.7 \pm 15.9$ | $\mathbf{84.2 \pm 10.5}$ | $82.3 \pm 15.7$ | 98.3 (Xu et al., 2023) |
| JSRT | $80.9 \pm 12.1$ | $\mathbf{88.4 \pm 6.8}$ | $84.7 \pm 11.3$ | 97.9(Liu et al., 2022) |
| Montgomery | $77.3 \pm 8.8$ | $78.3 \pm 8.6$ | $\mathbf{87.1 \pm 3.4}$ | 97.7(Liu et al., 2022) |
| Combined | $80.3 \pm 14.2$ | $\mathbf{84.4 \pm 9.9}$ | $83.6 \pm 13.6$ | - |

for DiNOv1-Diffusion and $N_{real} = 500$ for DiNOv2-Diffusion. Finally, DiNOv1-Diffusion yielded larger performance improvements and statistical significance when compared to DiNOv2-Diffusion.

**Zero-Shot Segmentation**: the performance of the zero-shot experiments are shown in Table 2. Both DiNOv1- and DiNOv2-Diffusion showed improvements of up to 10% Dice score when compared to a vanilla SD v1.5 model while also presenting lower variance. When addressing individual results, DiNOv1-Diffusion generated the best average Dice scores. Performance varied between datasets, with Montgomery (Jaeger et al., 2014) producing the lowest Dice scores for both vanilla Stable Diffusion and DiNOv1-Diffusion, but to the highest scores for the DiNOv2-based approach when comparing the overall best model. It should be noted that the best model checkpoint for segmentation was significantly earlier than the one found in Section 3. Moreover, the optimal parameters for DiffSeg were very similar for both self-supervised DMs, while the optimal merging threshold was 10x larger for the base SD model. Finally, non-optimal combinations of parameters produced significant artifacts in the generated masks as shown in Figure 5 (b). Additional zero-shot segmentation examples are provided in Section B.1 and an supplementary segmentation performance evaluation across different model checkpoints can be found in Section B.2.

## 4 DISCUSSION

DMs are a cornerstone in modern foundation models, revolutionizing many tasks in Computer Vision. Their ability to generate high-quality images has caused a large scientific, economic and societal disruption, whose long-term repercussions are difficult to foresee (Liu et al., 2024). However, despite their scientific and industrial utility, applying this technology in medical imaging is severely limited by key challenges such as a lack of large-scale labeled datasets including high-quality textual or non-textual descriptions (Kazerouni et al., 2023). Although this limitation might be temporary due to current trends in AI data acquisition and improved dataset interoperability (Akhtar et al., 2024), it is not clear whether the prevalent text-to-image generative recipe (Rombach et al., 2022) is optimal for medical applications.

Some approaches employing DMs in medical data exist. Chambon *et al.* (Chambon et al., 2022a) trained an SD architecture on the MIMIC-CXR dataset (Johnson et al., 2019) with good synthesis fidelity, reporting low FID scores and high accuracy scores on several downstream tasks including classification, report generation and image retrieval. However, their approach is severely limited on the size of the development dataset (300k images) and the low quality of accompanying captions. In histopathology, multiple authors have proposed applying DMs for image generation (Ye et al., 2023; Osorio et al., 2024). For instance, Aversa *et al.* relied on a custom-annotated dataset of large histopathology slides with segmentation masks representing different tissue subtypes within the slide and employed timestep unravelling to generate images larger than the typical 512px$^2$. However, their approach heavily relied on a closed-source, custom-annotated dataset, and timestep unraveling might be impractical in other medical imaging modalities. In contrast, Xu *et al.* (Xu et al., 2024) take a similar approach as the one proposed here, and train a DM conditioned only on an image encoder's $c_{CLS}$ for histopathology image synthesis. However, their method was partially supervised, as it relied on training additional label-specific DMs for $c_{CLS}$ generation. Besides being compute intensive, their method fails to leverage the emerging data augmentation and segmentation capabilities that a self-supervision DM training conveys. Finally, Pinaya *et al.* (Pinaya et al., 2022) trained an LDM on a large dataset of 31740 3D Brain MRI images from UK BioBank. However,

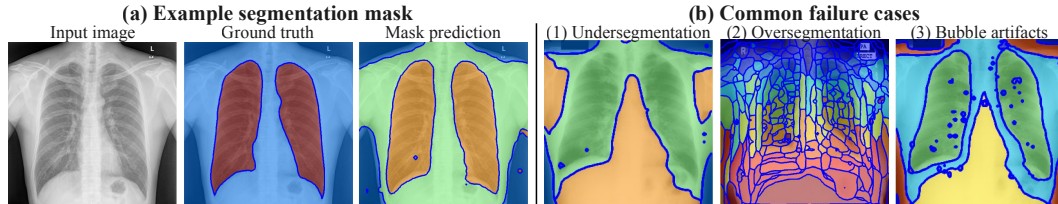

Figure 5: (a) Example segmentation masks generated by the best DiNOv1-Diffusion model and (b) common failure cases. Failures are caused by sub-optimal hyperparameters: (1) incomplete segmentation, often observed in early checkpoints or high thresholds; (2) oversegmentation, usually due to low merge thresholds; (3) bubble-like artifacts, mostly observed in later checkpoints.

despite the scale of this dataset, the fragmentation of clinical labels forced the authors to condition the DM with simplified clinical variables such as age, sex, ventricular volume, and brain volume.

DiNO-Diffusion addresses the data limitations in medical imaging by conditioning the image generation process on the images themselves. This allows training DMs on unlabelled data, which is more abundant in the medical field. The resulting DiNO-Diffusion models demonstrated good manifold coverage, as indicated by low FID scores, and exhibited notable properties in several downstream tasks. Firstly, adding synthetic data using the "reconstruction" strategy improved performance across most configurations. However, performance gains diminished as more real data became available, which is to be expected. Secondly, the "interpolation" strategy degraded performance in higher data regimes. We hypothesize that, although the generated images qualitatively resemble plausible images (see Figure 2-b), naïvely interpolating embeddings did not ensure that the interpolated labels corresponded to the decoded image's features, thereby hurting classification performance. We leave to future work the exploration of more sophisticated interpolation strategies. Thirdly, full synthetic training demonstrated that synthetic data can replace real data while preserving privacy, and even improve performance in small-data regimes, when used in abundance. Finally, DiNO-Diffusion's zero-shot segmentation outperformed a vanilla SD architecture. This is remarkable given that the dataset used to train the vanilla SD model was several orders of magnitude larger. Despite DiNO-Diffusion's performance, conditioning the synthesis process on image embeddings has theoretical advantages and disadvantages. This type of conditioning relaxes the need for annotations, enabling the collection of larger datasets for model training, and has proven effective across various tasks. However, usage of an image-conditioned model is fundamentally different from text-based approaches, as image generation requires conditioning on an image. Still, this circular dependency between input and output could be advantageous in some use cases, such as data augmentation or privacy-preserving data sharing.

These advantages and disadvantages evidence room for improvement. Firstly, DiNOv1-Diffusion outperformed DiNOv2-Diffusion both quantitatively and qualitatively, despite the larger data pool used to train the DiNOv2 image encoder (Oquab et al., 2023). This suggests that using domain-specific encoders (Cohen et al., 2022; Pérez-García et al., 2024; Moutakanni et al., 2024), or even a combination of different image encoders (Esser et al., 2024; Liu et al., 2024) could further improve these results. Secondly, DiNO-Diffusion would benefit from more recent diffusion architectures found in the literature (Esser et al., 2024; Liu et al., 2024; Betker et al., 2023). Thirdly, generation based on other descriptors, such as text, could be enabled by using external networks to map the image embedding space to the text embedding (Zhang et al., 2023b; Li et al., 2023). Finally, the failure cases found in the zero-shot segmentation workstream require adapting the DiffSeg methodology to datasets with different characteristics, including image-level hyperparameter optimization, further attention-merging strategies, or using DiNO's attention maps to better locate anatomic structures.

In conclusion, while diffusion models have significantly impacted the Computer Vision community with broad scientific, economic, and societal implications, their application to medical imaging is constrained by data and annotation scarcity. Our DiNO-Diffusion approach addresses this problem by conditioning the image generation on the images themselves, eliminating the need for extensive annotations. The approach shows promising results in manifold coverage, data augmentation, privacy preservation and zero-shot segmentation. Finally, this work underscores the need for innovative solutions in medical imaging to fully leverage the potential of DMs in this space.

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

APPENDIX

# A   IMAGE GENERATION

## A.1   RECONSTRUCTIONS FROM ENTIRE DINO EMBEDDINGS

Initial explorations showed that the DiNO encodings ($c$) need to be compressed before feeding them as conditioning during DiNO-Diffusion training. As depicted in Figure 6, when using the entire DiNO image embedding ($c$) for conditioning, the model can learn to utilize the information richness of the DiNO encoding to reconstruct the initial CXR image with exceptional detail. This richness can likely be attributed to the various patch tokens ($c_{LCL}$) that the whole DiNO encoding includes, since they retain local information about the corresponding image patch, but also how it relates to the remaining patches in the image. In addition, Figure 6 further shows how the information gathered in the DiNO encoding seems to be sufficient to allow our model to reconstruct images from modalities that have never been seen during the DiNO-Diffusion training (only CXR).

Further supporting this, 7 shows the reconstructions yielded by the model trained with patch tokens, when the condition has half the tokens from an image and half the tokens from another image. In this scenario, the regions corresponding to the patch tokens from Image A are reconstructed according to image A, while the equivalent occurs for the regions corresponding to patch tokens of Image B. The abrupt transition in the final image, clearly depicts how the model trained with patch tokens learns a one-to-one correspondence between each patch token and its corresponding image patch, as opposed to any global understanding of the underlying image domain.

Considering these two experiments would not be applicable for image data augmentation purposes. In this work, we show that the latter is only achievable if the conditional information is limited to the tokens that gather global information from the image ($c_{GLB}$).

## A.2   RECONSTRUCTION AND INTERPOLATION FROM NON-STANDARD IMAGES

DiNO-Diffusion yields high-quality reconstructions and interpolations, which we have shown to be applicable for data augmentation and possibly privacy preservation purposes. Nevertheless, when the base real images include non-standard elements, like ECG-electrodes, cables or support devices

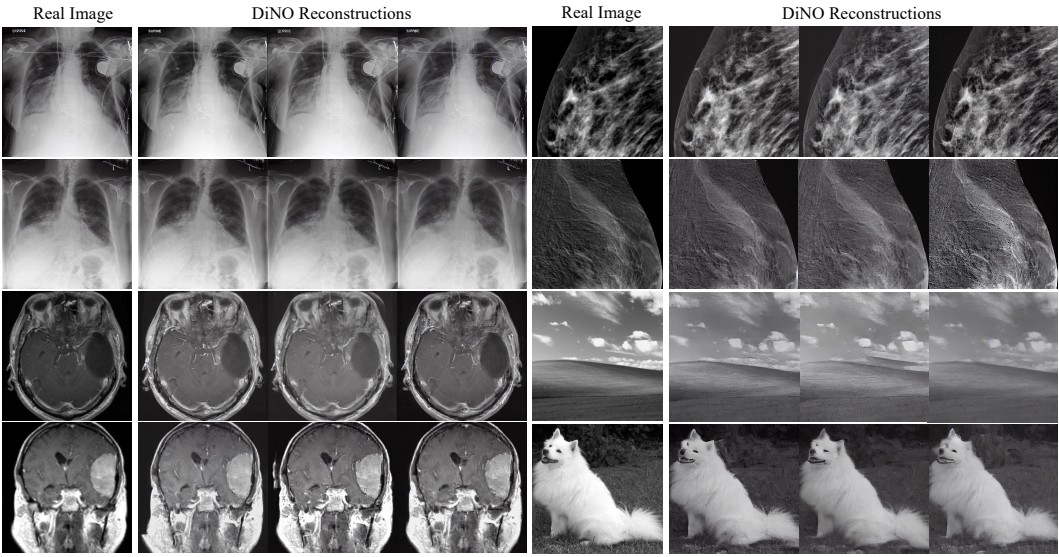

Figure 6: Examples of DiNOv1-Diffusion reconstructions when the training (and inference) is conditioned on the entire DiNO embedding ($c$) rather than just the global information ($c_{GLB}$). Specific checkpoint showed here is after $110k$ training steps. Note that the whole DiNO encoding seems to encode enough information to enable the reconstruction of images from modalities that have never been seen during the DiNO-Diffusion training (only CXR), like brain MRI, mammography and even natural imaging.

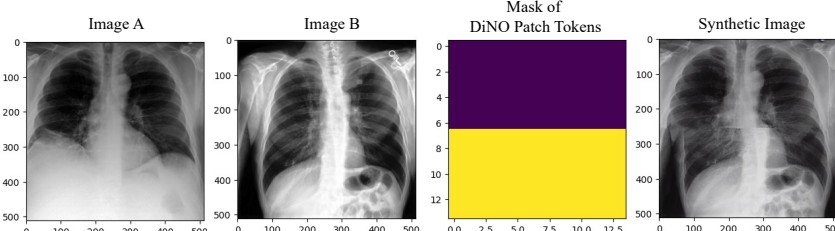

Figure 7: Examples of DiNOv1-Diffusion reconstructions when the training (and inference) is conditioned on the entire DiNO embedding ($c$) rather than just the global information ($c_{GLB}$). Specific checkpoint showed here is after $40k$ training steps. Here the inference uses a conditioning DiNO embedding with local patches deriving from two different image. As depicted by the mask, the first half of the patch tokens refer to Image A, while the second half to Image B. The final reconstruction is an image with an abrupt transition between accurate reconstructions of each half of Image A and B. This experiment evidences how this training setting leads to the model learning a simple one-to-one reconstruction of each patch token, as opposed to any global understanding of the underlying image space.

like pacemakers, the synthetic variants can become less realistic. Figure 8 highlights a few of these examples and their variants when using both DiNOv1 and DiNOv2. Particularly in Figure 8a, note how not only anatomical variance but also device location and morphological variance is introduced into the reconstructions. While some of these variants might retain high-quality, the extent to which this variance is still realistic can only be accurately determined by subject experts. We leave this supplementary study as future work.

Figure 9 presents a more extensive collection of both successfully and poorly segmented examples from multiple datasets and DiNO-Diffusion variants. There are apparent difference between the different datasets, with Montgomery in particular tending to be oversegmented, while Shenzhen and JSRT are undersegmented. This coincides with observations made about hyperparameters as shown in B.3, where the Montgomery dataset appears to require different hyperparameters for optimal performance. As shown in particular on the well-performing Montgomery-cases, both models produce good results even for deformed images. While the lung lobes can be segmented well, it appears that the heart is rarely clearly detected. Whether this is a result of higher difficulty of segmentation for the heart, as compared to the often clearly contrasted lung lobes, or an artifact of the training data, where the heart might often not be completely present, remains subject to further research.

## B  SEGMENTATION

### B.1  MORE SEGMENTATION EXAMPLES

### B.2  MODEL CHECKPOINTS

Segmentation performance for both models, as shown in Figure 10, consistently peaks relatively early during training. This indicates that reconstructive performance indicated by FID-score is not strongly correlated to segmentation capabilities. The DiNOv1-based model reaches peak performance after less than $20k$ steps, while DiNOv2 reaches highest average Dice-score slightly later, between $20k$ and $30k$ steps. This difference in best observed performance, albeit much earlier during training, is consistent with the observed progression of FID scores as seen in Figure 3. The performative decrease slows down around $60k$ steps. There are notable differences in results between the three datasets. For both models, evaluation on the Montgomery leads to highest scores earlier in training than for the other two models. This is possibly explained by domain specific variance between the datasets, where longer training fits the JSRT and Shenzhen dataset distribution better than that of Montgomery. While peak performance on Montgomery equals or even exceed that of the other datasets, it's lower bound also appears much lower than the other models' bounds, further suggesting a dataset-specific difference.

Figure 10 also reveals another notable difference between datasets: both JSRT and Shenzhen show a significant difference between median and mean performance, with the latter being lower across most timesteps. This suggests a higher variance in image-level scores with a larger amount of good

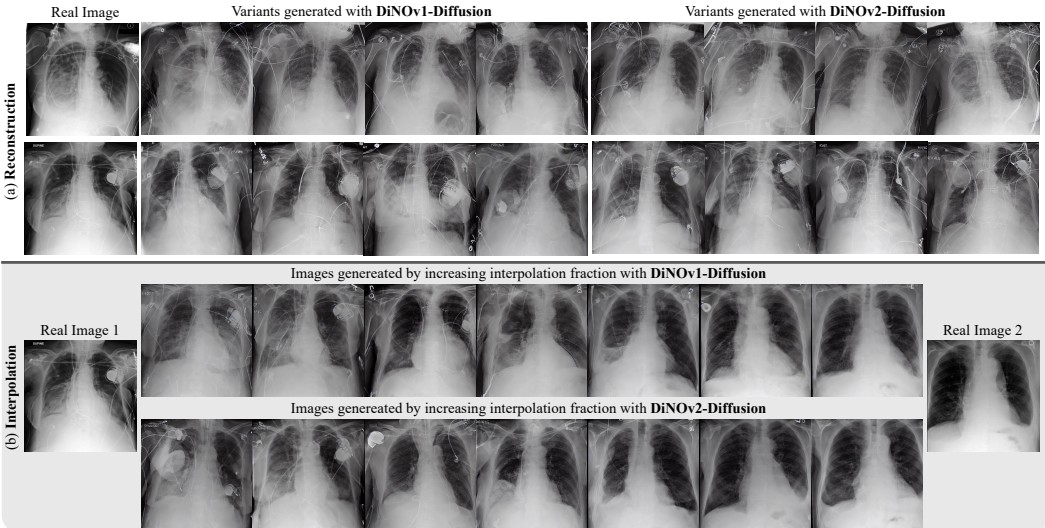

Figure 8: Failure cases of generated images with DiNO-Diffusion. In the reconstruction experiment (a), each row represents randomly generated examples from two base images within MIMIC and for both DiNOv1-Diffusion and DiNOv2-Diffusion, showing semantic anatomical variability but faulty reconstructions of the pacemaker and ECG electrodes. In the interpolation experiment (b), each row depicts two real images and the result from generating synthetic images by interpolating the embeddings incrementally for the DiNOv1-Diffusion (b-top) and DiNOv2-Diffusion (b-bottom) settings. The sampling between the the two examples is smooth but reconstructions closer to the image with the pacemaker look less realistic.

and bad outliers. Conversely, mean and dice scores for both models on Montgomery are very similar across all checkpoints. A possible explanation is a potential larger homogeneity of images within Montgomery, leading to a more narrow distribution of Dice scores. Because failed segmentation on Montgomery at the overall optimal HP-configuration tends to be caused by oversegmentation, as shown in Figure 9, it is also conceivable that the Dice metric punishes these cases less harshly, leading to more consistent scores. A further investigation into optimal metrics is warranted to confirm this thesis.

## B.3 SEGMENTATION HYPERPARAMETER EVALUATION

We show the segmentation performance for the two most relevant hyperparameters, merge threshold and timestep for both models to select the optimal configuration as done in (Tian et al., 2023). Other hyperparameters (anchor grid size, clustering based refinement) did not show meaningful performance differences across a reasonable range of values and were kept at the default values.

As shown in Figure 11, the optimal merging threshold for both models can be found around 0.05 for the average dataset. Analysis of the individual datasets reveals large differences. Particularly the Montgomery dataset produces better results at higher thresholds for both DinoV1 and DinoV2. Conversely, JSRT requires a slightly lower threshold for optimal performance. Shenzhen matches the average performance, which could be partially explained by it's size, as it makes up the largest portion of the three datasets. Understanding which specific image / dataset characteristic could better inform the optimal threshold remains an interesting question.

Finally, different timesteps were evaluated in Figure 12. Results in very early steps (closer to 1000) lead to bad results for all datasets. Optimal performance was achieved on a relatively large plateau between timestep 100 to 500 and degrades sharply after timestep 600. This is in line with the results presented in (Tian et al., 2023). As with the merge threshold, there are dataset-specific differences. The Montgomery dataset leads to better results at higher timesteps, while JSRT peaks rather early.

The results of hyperparameter-tuning highlight the importance of careful parameters selection. Different datasets or data-distributions within the segmented data can require different hyperparameters. A thorough investigation of dataset or even image-specific parameters detection poses a topic for further research.

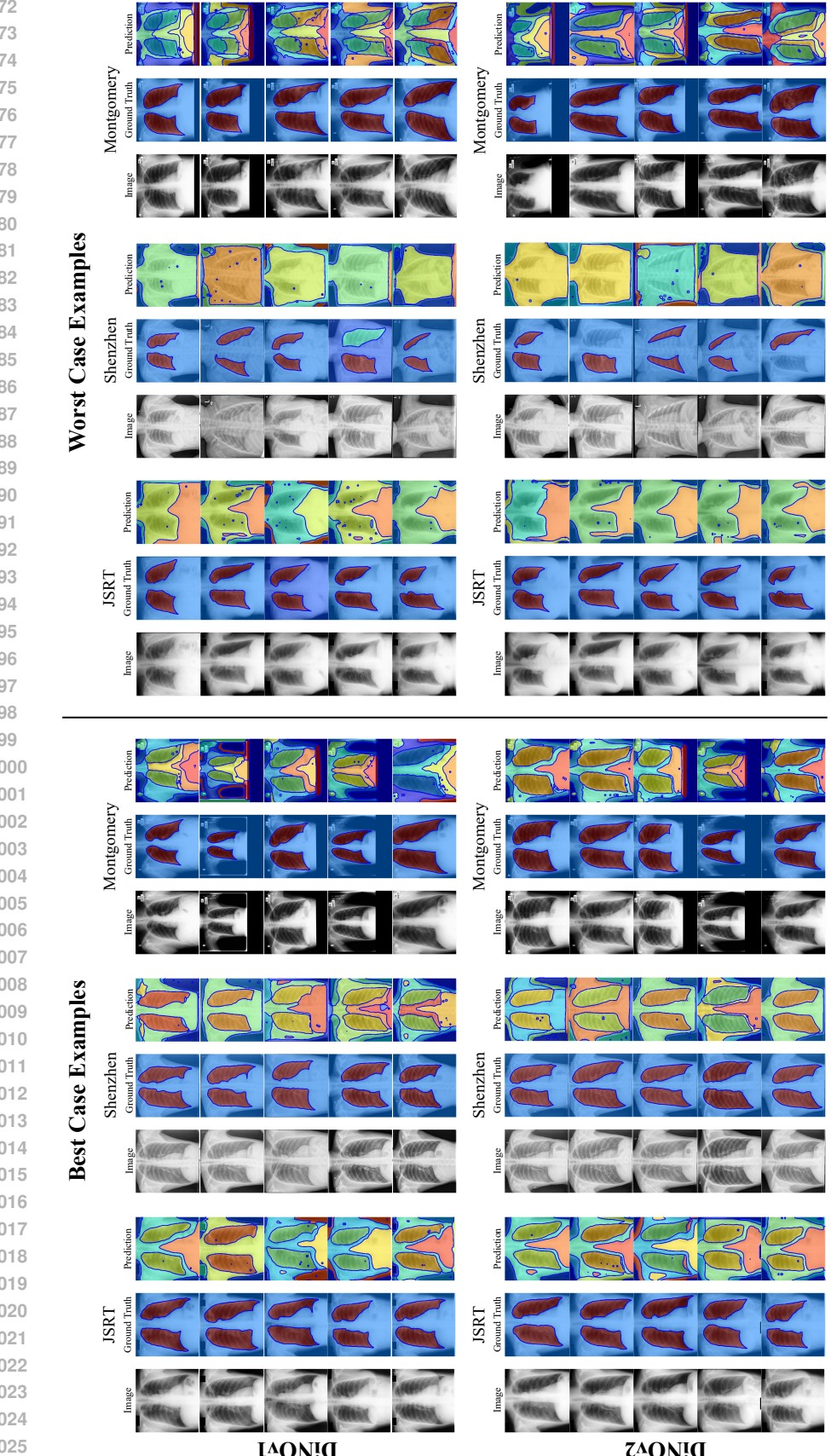

Figure 9: Examples of masks generated on all three datasets with the two models. Both models appear to be robust to deformations in the input images, as demonstrated by good performance particularly on the Montgomery cases. Even in good cases, we can observe "bubble"-artifacts, which could potentially be a result of overfitting and strong local attention. Whether failure cases are over-or undersegmented appears to be dataset-dependent: we find that both JSRT and Shenzhen show undersegmentation in the failure cases, whereas Montgomery tends to be oversegmented. This is likely caused by a shift in domain-specific image features (such as saturation and dynamic range) and highlights the potential for further improvement via image-or dataset specific hyperparameter tuning.

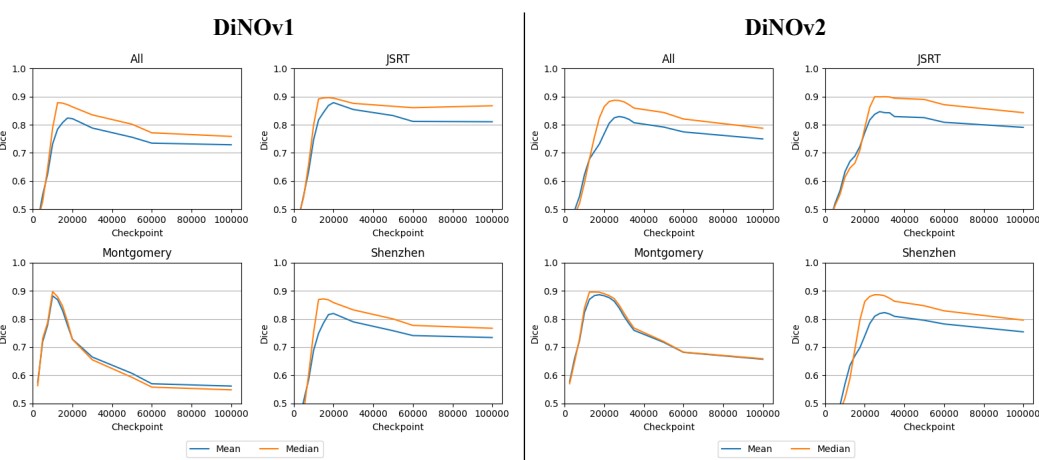

Figure 10: Segmentation performance across model checkpoints.

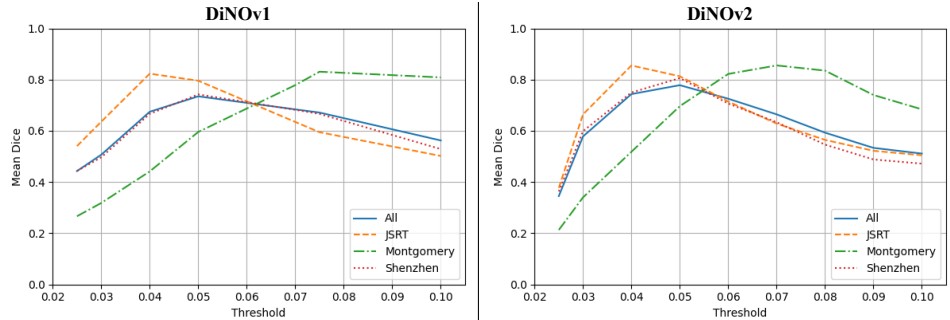

Figure 11: Tuning of merging threshold across datasets. All results were produced at the same checkpoint and timestep.

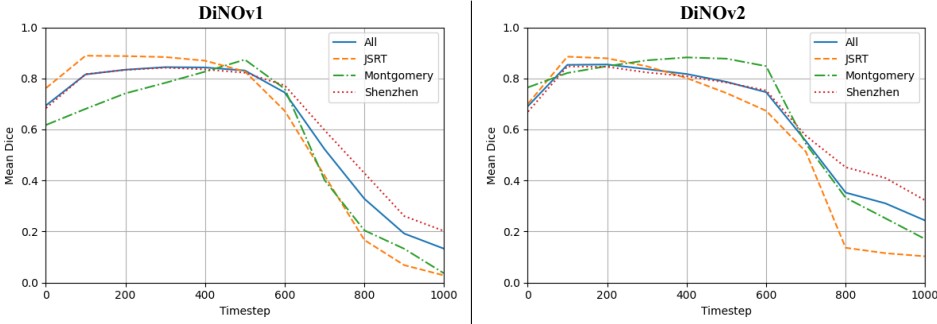

Figure 12: Tuning of timestep across datasets. All results were produced at the same checkpoint and threshold

