# OpenReview forum: "DiNO-Diffusion: Scaling Medical Diffusion Models via Self-Supervised Pre-Training"
_ICLR.cc/2025/Conference — ICLR 2025 Conference Withdrawn Submission_

### Official Review · Reviewer_r5X8 · 2024-10-15

**Soundness:** 3
**Presentation:** 3
**Contribution:** 3
**Rating:** 8
**Confidence:** 4

**Summary:**

This paper presents DiNO-Diffusion, a self-supervised method designed to train Diffusion Models (DMs) for medical imaging without the need for large annotated datasets. Traditional DMs require extensive labeled data, which is often scarce in medical applications. DiNO-Diffusion addresses this limitation by conditioning the generation process on image embeddings extracted from DiNO, a pretrained vision transformer, allowing the use of over 868k unlabeled chest X-Ray (CXR) images from public datasets.

**Strengths:**

Originality:
1. Self-Supervised Training: DiNO-Diffusion introduces a novel approach for training diffusion models without large annotated datasets, addressing a key limitation in medical imaging.
2. Creative Integration: Combines diffusion models with image embeddings from DiNO, a pretrained vision transformer, showcasing an innovative fusion of advanced techniques.

Quality:
1. Robust Experimental Design: Utilizes over 868k unlabeled chest X-Ray (CXR) images, ensuring the method's scalability and reliability.
Comprehensive Evaluation: Assesses performance using FID scores, AUC improvements, and Dice scores, providing a thorough validation of the approach.
2. Strong Results: Demonstrates significant enhancements in classification (up to 20% AUC increase) and impressive zero-shot segmentation (up to 84.4% Dice score), highlighting effectiveness.

Clarity:
1. Well-Organized Structure: Clearly structured sections on methodology, experiments, and results facilitate easy understanding.
Detailed Explanations: Thoroughly explains key components like DiNO embeddings and their integration with diffusion models.
2. Effective Visuals: Uses visual aids to illustrate qualitative comparisons and segmentation outcomes, enhancing comprehension.

Significance:
1. Overcoming Data Scarcity: Enables training of diffusion models without extensive annotations, broadening their applicability in medical imaging.
2. Enhancing Downstream Tasks: Improves data augmentation, leading to significant gains in classification and segmentation performance.
3. Scalability and Adaptability: Easily adaptable to other medical imaging modalities and compatible with state-of-the-art diffusion models, supporting large-scale, multi-domain applications.

**Weaknesses:**

1. My main concern for the medical segmentation task is whether the performance on zero-shot segmentation can exceed the current leading methods like MEDSAM or MEDSAM2.
2. I am curious if the dataset will be made available to the community.
3. It would be better to define DiNOv1-Diffusion and DiNOv2-Diffusion in the caption of the first figure.

**Questions:**

Extensive experiments on the large dataset make this work a solid submission. I have no questions except those mentioned in the weaknesses.

---

### Official Review · Reviewer_sUX6 · 2024-10-31

**Soundness:** 2
**Presentation:** 2
**Contribution:** 1
**Rating:** 3
**Confidence:** 5

**Summary:**

This paper proposes the DiNO-Diffusion method, which uses self-supervised image representation to guide diffusion for chest X-ray images and evaluates the diffusion model on three tasks: reconstruction, interpolation, and zero-shot lung lobe segmentation.

**Strengths:**

The paper is generally clear and easy to understand.

**Weaknesses:**

1. The paper does not demonstrate the value of DiNO-Diffusion in the medical domain. The three evaluation methods mentioned—reconstruction, interpolation, and zero-shot lung lobe segmentation—do not seem to hold significant value for the medical field, in my understanding. In the context of X-ray scanning, what clinicians are more concerned about is the ability to detect lesions from images, particularly those that current X-ray models cannot address, but which could potentially be solved using DiNO-Diffusion. Critical evaluations and analyses are lacking in this regard.

2. The novelty of the method is limited. There are similar works that use self-supervised models to guide diffusion [1,2,3]; the basic idea is typically to use self-supervised image embedding as a condition or to utilize self-supervised representation for clustering to generate pseudo labels as conditions. The method presented in this paper falls into this category but does not directly reference these works or perform comparisons with them.

[1] Vincent et al., Self-Guided Diffusion Models, CVPR 2023

[2] Vincent et al., Guided diffusion from self-supervised diffusion features, 2023.

[3] Alexandros et al., Learned representation-guided diffusion models for large-image generation, CVPR 2024.

3. The related work section is insufficient and lacks a detailed survey and comparison of works that guide diffusion with self-supervised methods.

**Questions:**

Please see Sec. Weakness for details.

---

### Official Review · Reviewer_xuVX · 2024-11-02

**Soundness:** 3
**Presentation:** 4
**Contribution:** 2
**Rating:** 3
**Confidence:** 4

**Summary:**

The paper proposes DiNO-Diffusion models, which leverage embeddings generated by a self-supervised transformer trained with DiNO methods. The approach is evaluated on the MIMIC-CXR dataset for three tasks: reconstruction, interpolation, and segmentation.

**Strengths:**

1. The motivation of the paper is clear, aiming to facilitate the training of diffusion models for medical dataset generation in sparsely-annotated settings.

2. It is a good idea to apply self-supervised learning for diffusion model training.

3. The results are well-presented.

**Weaknesses:**

1. The structure of DiNO-Diffusion is similar to latent diffusion models [1], where the conditional latent features extracted from an encoder are injected at each denoising step. It will be interesting to include latent diffusion models as the baseline to compare.

2. Although the paper conducts 3 types of evaluations, only one dataset (MIMIC-CXR) serves as the testbed. In order to demonstrate the better generalization, the paper can be more strengthful by adding additional datasets. For example, including other chest X-ray datasets like CheXpert or MIMIC-CXR can be options.

3. As a conditional diffusion model, it seems like the paper does not include sufficient baseline methods for comparison, for example, ControlNet[2] and SegGuidedDiff [3]. These methods can generate images given input conditions, and can be finetuned in an end-to-end manner. It would be beneficial for the authors to discuss in the paper why leveraging features extracted from self-supervised model might outperform end-to-end finetuning approaches.

[1] Rombach, Robin, et al. "High-resolution image synthesis with latent diffusion models." Proceedings of the IEEE/CVF conference on computer vision and pattern recognition. 2022.

[2] Zhang, Lvmin, Anyi Rao, and Maneesh Agrawala. "Adding conditional control to text-to-image diffusion models." Proceedings of the IEEE/CVF International Conference on Computer Vision. 2023.

[3] Konz, Nicholas, et al. "Anatomically-controllable medical image generation with segmentation-guided diffusion models." International Conference on Medical Image Computing and Computer-Assisted Intervention. Cham: Springer Nature Switzerland, 2024.

**Questions:**

Including a self-supervised encoder to assist medical image generation is a good idea. However, the training of latent diffusion model [1] can also leverage no annotated data. In addition, the latent diffusion model can incorporate multiple conditions such as text, images and labels. In my opinion, it would be better for the author to demonstrate the embeddings generated by the self-supervised encoder were more informative and powerful to guide the image generation. Specifically, I would suggest conducting experiments that directly compare downstream tasks performance (e.g., the segmentation) between using DiNO embeddings and others, like embeddings extracted from the  encoder of latent diffusion models.

---

### Official Review · Reviewer_fGGi · 2024-11-03

**Soundness:** 3
**Presentation:** 2
**Contribution:** 3
**Rating:** 5
**Confidence:** 4

**Summary:**

This paper presents a diffusion model designed to generate synthetic images for pre-training AI, which can supplement real images in downstream tasks. The proposed DiNO-Diffusion model offers the advantage of conditioning image generation on the images themselves, using features extracted by DiNO. Experimental results indicate that synthetic images provide effective data augmentation and hold potential for zero-shot segmentation.

**Strengths:**

+ The ablation study is thorough, with Table 1 examining multiple configurations of the proposed method. The comparisons among V1/V2, reconstruction/interpolation, and different rs ratios provide valuable insights.

+ The authors include numerous qualitative visualizations, clearly illustrating the outputs of reconstruction- and interpolation-based methods for readers.

+ The analysis of failure cases in Figure 5 is crucial for identifying the method's weaknesses and exploring areas for improvement.

**Weaknesses:**

- The novelty of the method is a concern, as there are no substantial modifications to Stable Diffusion. This is not necessarily a weakness if Stable Diffusion already performs the task effectively. However, since the authors introduce DiNO to the diffusion model (see title), it would be expected that they explain how DiNO is incorporated into Stable Diffusion. Yet, this integration is not shown in Figure 1 and is only briefly mentioned in the methods section.

- It is unclear why the generated synthetic images are considered semantically diverse. The generated images rely solely on image features computed from the existing images in the training set or interpolated features from these images. There is no assurance that the generated images are out-of-distribution or that they can be controlled by interpretable features, as text-based conditioning would allow.

- The proposed experimental settings—data augmentation and full synthetic training—are fundamentally similar. In the paper, data augmentation uses both real and synthetic images, while full synthetic training relies solely on synthetic images. However, since training the diffusion model (DM) requires real images, full synthetic training is not completely independent of real data; it merely uses the DM to encode information from the real data, allowing real images to be omitted during "full synthetic training." Consequently, the results of these two settings are expected to be similar, as reported in the paper.

- It is unclear how synthetic data can be used for downstream task training. Since downstream task training relies on supervised learning, how are annotations generated for the synthetic data?

**Questions:**

1. How can synthetic images with semantic labels be generated for downstream tasks? From Figure 1, it seems that synthetic images cannot be generated with semantically meaningful conditions. The DM is conditioned on an image descriptor, which lacks semantics if the image is unannotated.

2. In Figure 1, I don’t see any references to DiNO. How is DiNO incorporated into the proposed framework?

3. In Figure 1b (i), the reconstructed image appears significantly different from the input image. How does the reconstruction network generate a horizontally flipped image?

4. In Figure 2, the generated images, both reconstructed and interpolated, have lower intensity (appear darker) than real images. What is causing this? Are these images generated by DM using reconstruction and interpolation features clinically meaningful?

5. I don’t see a clear distinction between data augmentation and fully synthetic training. Both approaches require real data to train the DM, so real data is utilized in both cases. Therefore, their results should theoretically be similar.

6. Using image embeddings as conditions is novel, but how does it improve interpretability compared to using text embeddings? Image embeddings are derived from existing real images, and obtaining embeddings for unseen or out-of-distribution images is challenging (not mentioned in the paper). Additionally, it's unclear if mix-match image features is truly meaningful, as it is with text conditioning.

---

### Note · Authors · 2024-11-25

**Comment:**

We thank all the reviewers for reading and providing insightful feedback on our work. Unfortunately, due to internal circunstances, we currently do not possess the bandwidth for addressing all the points raised by the reviewing team. We thank them for their effort but, after careful consideration, we decided to withdraw our submission.

**Withdrawal Confirmation:**

I have read and agree with the venue's withdrawal policy on behalf of myself and my co-authors.